# WiTransformer: A Novel Robust Gesture Recognition Sensing Model with WiFi

**DOI:** 10.3390/s23052612

**Published:** 2023-02-27

**Authors:** Mingze Yang, Hai Zhu, Runzhe Zhu, Fei Wu, Ling Yin, Yuncheng Yang

**Affiliations:** School of Electronic and Electrical Engineering, Shanghai University of Engineering Science, Shanghai 201602, China

**Keywords:** body-coordinate velocity profile, channel state information, human activity recognition, transformer, WiFi signals

## Abstract

The past decade has demonstrated the potential of human activity recognition (HAR) with WiFi signals owing to non-invasiveness and ubiquity. Previous research has largely concentrated on enhancing precision through sophisticated models. However, the complexity of recognition tasks has been largely neglected. Thus, the performance of the HAR system is markedly diminished when tasked with increasing complexities, such as a larger classification number, the confusion of similar actions, and signal distortion To address this issue, we eliminated conventional convolutional and recurrent backbones and proposed WiTransformer, a novel tactic based on pure Transformers. Nevertheless, Transformer-like models are typically suited to large-scale datasets as pretraining models, according to the experience of the Vision Transformer. Therefore, we adopted the Body-coordinate Velocity Profile, a cross-domain WiFi signal feature derived from the channel state information, to reduce the threshold of the Transformers. Based on this, we propose two modified transformer architectures, united spatiotemporal Transformer (UST) and separated spatiotemporal Transformer (SST) to realize WiFi-based human gesture recognition models with task robustness. SST intuitively extracts spatial and temporal data features using two encoders, respectively. By contrast, UST can extract the same three-dimensional features with only a one-dimensional encoder, owing to its well-designed structure. We evaluated SST and UST on four designed task datasets (TDSs) with varying task complexities. The experimental results demonstrate that UST has achieved recognition accuracy of 86.16% on the most complex task dataset TDSs-22, outperforming the other popular backbones. Simultaneously, the accuracy decreases by at most 3.18% when the task complexity increases from TDSs-6 to TDSs-22, which is 0.14–0.2 times that of others. However, as predicted and analyzed, SST fails because of excessive lack of inductive bias and the limited scale of the training data.

## 1. Introduction

Human activity perception and state monitoring based on wireless sensing are crucial components in the development of healthcare [1,2,3], human-computer interaction [4], and smart living [3]. Initially, intuitive methods based on computer vision [4] and wearable sensors [5] were employed. However, computer vision has issues including privacy leakage and lighting conditions and wearable devices require users to carry additional devices. These limitations have prompted the development of new methods. Based on the fact that WiFi wireless radio waves will be reflected when encountering obstacles (as shown in Figure 1) [6], human activity recognition (HAR) based on commercial off-the-shelf (COTS) WiFi has become a new research focus because of its cost-effectiveness, user privacy leakage, ubiquity, and passive user perception nature. Specifically, WiFi signals that can be used to identify and perceive HAR generally includes the received signal strength indication (RSSI) and channel ctate information (CSI). RSSI is easy to obtain but coarse-grained. It is suitable for coarse-grained indoor positioning [7] and behavior recognition [8]. By contrast, CSI is a type of stable, fine-grained information. Thus, it has become an essential signal feature for motion tracking [9,10], centimeter-level positioning [11], and human gesture recognition (HGR) [12].

CSI-based WiFi wireless sensing approaches are typically divided into two categories. One is driven by expert knowledge and based on physical models [12,13,14], and the other is driven by big data and based on machine learning (ML) models. The former attempts to implement a type of sensing system using an interpretable and analytical mathematical model relying on physical principles, whereas the latter collects a large amount of CSI data and extracts their physical or statistical features to train ML models for HAR. Among these related works, CARM [6] is a pioneering research work in which the hidden Markov mode l (HMM) was used to model the spectral features of WiFi, and nine types of daily behavior recognition were realized. However, these early systems might neglect the specificity of objects, the environmental dependence of signal features, and the intricacy of recognition tasks.

With the maturity of neural networks, research using data-driven methods is increasingly adopting deep learning (DL) models, such as convolutional neural networks (CNNs) and recurrent neural networks (RNNs). They are used to mine high-dimensional data features to enhance the robustness of a system and further improve its recognition accuracy [15,16,17]. LCED [17] uses a long short-term memory neural network (LSTM) to encode CSI time-series information after preprocessing and utilizes a CNN model as a decoder to implement a type of user robust recognition system. The methods that rely solely on neural network modeling to enhance system generalization might not consider the characteristics of WiFi signals. As a result, their performance is limited when facing complex cross-environment tasks.

However, when considering deployment in a real environment, DL-based ways face several challenges. In the first place, compared with cameras in CV heavily relying on DL, the ubiquity of WiFi devices is limited and inadequate. Thus, collecting large-scale cross-domain data for training models is challenging. It is difficult to eliminate the dependence on domains (such as location, orientation, and user identity) by simply training models without big data. To address this issue, researchers have conducted research and made improvements from the bottom (signal level) to the top (model level) of sensing systems [13,18,19]. Widar3.0 [13] first solved the problem using a model combining CNN with RNN (CRNN) and body-coordinate velocity profile (BVP), a domain-independent signal feature derived from CSI. Although Widar 3.0 theoretically overcomes the generalization of the recognition systems at the signal feature level, it relies heavily on expert experience and knowledge. WiGr [19] is one of the most advanced and latest research works, and only a pair of WiFi can be used to enhance the generalization of the system at the model level. Compared with Widar 3.0, it still requires secondary training, although it requires an excessively amount of data for training.

However, the complexity of a recognition task also affects the performance of the system, such as the classification number (**Classification**), the distortion caused by part information loss (**Distortion**), and similarity in the performance path (**Similarity**). RNN and CNN, lacking long-range dependence and global feature capture, present difficulty in harnessing redundant data information to resist complexity changes among varying recognition tasks. Widar3.0 can achieve approximately 90% recognition accuracy across multiple domains for the six types of gestures. However, the performance decreased by approximately 20% when we test their model on a dataset with multiple task complexities. After discussion and analysis, we identified three issues regarding the results. First, CNN cannot obtain correlations among global features, and GRU (RNN-like models) cannot capture long-range dependencies owing to memory limitations. These defects restrict the expressive (fitting) capability of the CRNN [20]. Therefore, the model was affected when there were more classifications. Second, unlike digital images, a BVP sequence comprises a series of sparse matrices with only a little redundant information. Meanwhile, it is more difficult for WiFi devices to obtain stable signals than optical devices. Even if filters exist, signal aberrations introduced by overlapping limbs are common and unavoidable. However, the recognition is prone to frequent interference. Thus, long-range redundant spatial information scarcely helps the model resist signal distortion under CNN for BVPs. Third, the signal power distribution caused by gestures is concentrated. If distant spatial features cannot be considered, it will be difficult for the model to distinguish between similar gestures. In contrast, the Transformer [21] and its Self-Attention can take full advantage of information redundancy to address these problems [22]. The transformer encoder is a long-range feature extractor. Its architecture and Attention can take full advantage of redundant information to deal with the above problems. Inspired by ViT, we learned that backbone networks, such as convolution and recursion, are unnecessary for Attention, which is the foundation of our innovation.

The Transformer has become a significant model framework in the natural language processing (NLP) and computer vision (CV) fields. However its application in WiFi wireless sensing remains in its infancy. In existing research, the Transformer or its self-attention is either combined with other backbone networks [22,23,24,25] or used to replace some components of other backbone networks without changing the overall structure of the network [15,26]. However, these studies do not consider the use of Transformers that capture global features and redundant information to resist the complexity of recognition tasks.

We believe that, as long-range and global feature extractors, transformer encoders can leverage redundant information to handle these three issues mentioned above well. Furthermore, inspired by ViT, we learned that backbone networks such as convolution and recursion are unnecessary. Therefore, in this study, we explore and study how to help WiFi-based HGR resist the complexity change of recognition tasks using a pure transformer encoder and the domain-independent signal feature, BVP. To the best of our knowledge, this is the first study to implement HGR using WiFi signals with only Transformers. The main contributions of this study are summarized as follows:To improve the identification of spatiotemporal features and enhance the robustness for recognition task complexity changes, we explore and design two types of three-dimensional (3D) spatiotemporal feature extractors, the united spatiotemporal transformer model (UST) and the separated spatiotemporal transformer model (SST), based on pure modified transformer encoders;For UST, we propose a 3D spatiotemporal feature fusion method, stacking-fusion, and tube embedding, which fully simplifies the Transformer-like spatiotemporal fusion architecture and reduces the time complexity. Moreover, the 3D spatiotemporal feature fusion and modeling are realized only by a one-dimensional (1D) encoder;We also propose a novel 3D time-series marking strategy, channel position embedding (CPE), for UST to supplement the absence of position information between different BVP input frames;We conducted experiments on regrouped Widar 3.0 public datasets. The results demonstrated that the UST of WiTransformer, as a category of the backbone network, has higher recognition accuracy and better robustness against the task complexity change compared with other popular models with the ability of 3D-feature extraction (e.g., Conv3D and CNN+GRU). In addition, for the task containing simultaneously three types of task complexity, UST also achieves 89.34% average recognition accuracy outperforming state-of-the-art works (including Widar 3.0, THAT, LCED, and CARM).

## 2. Related Works

The detection, extraction, and recognition of dynamic and static objects in an image or video using feature extractors have been mature in the traditional field of CV. Wiseman et al. [27] and M.P. et al. [28] pioneered research into static and dynamic contour detection and extraction in images and videos, respectively. Recently, researchers have further developed advanced automatic feature extractors and target segmentation techniques to enhance the perceptual boundaries of optical sensors [29,30]. However, WiFi signals cannot be modeled by visible objects, but rely more on the spectral features of signals to reproduce the characteristics of objects, which is unlike CV.

Based on the multipath effect, the WiFi signals from the transmitters to the receivers is affected by dynamic (such as the human body) and static (such as walls) objects. Using the microcomputers of the receivers, the dynamic patterns of the signal change caused by a human can be saved, as shown in Figure 1. A sensing scheme that utilizes these patterns depends on expert knowledge-driven or data-driven models to determine the mapping between signals and motions. Consequently, in this section, we first introduce some studies on these two aforementioned methods. Subsequently, regarding the DL approach as an outset, the Transformer and attention are presented in several related studies.

### 2.1. Expert Knowledge-Driven Method

The Fresnel zone theory, Doppler effect, and wave velocity are all natural physical laws for WiFi wireless signals. Under laws and empirical measurements, the links between the movements of human bodies and physical qualities caused by the movement can be found manually and visually by researchers, such as the Fresnel zone model [31], Doppler shift spectrum [9,11], angle of arrival (AOA) [32] and time of flight (TOF) [33]. By exploiting these links, analytical mathematical models can be constructed to realize motion tracking [9,11,33], state detection [31], and motion recognition [34]. We refer to these expert knowledge-driven methods.

However, expert knowledge-driven methods typically have three limitations. First, models require substantial manual labor and specialized knowledge for development. Second, owing to the complexity of the system, explicit deterministic models may miss implicit data features. Third, the analytical models may contain hyperparameters from unknown environmental factors that cannot be detected artificially.

### 2.2. Data-Driven Method

Compared to expert knowledge-driven methods, a data-driven method can extract high-dimensional data features that humans cannot capture based on information theory and statistical principles. The former lacks flexibility and adequate recognition accuracy and is labor-intensive and time-consuming.

Initially, Wang et al. [6,35] creatively regarded the speed of body movement as a quantitative indicator connecting human activities and CSI dynamic patterns based on the exploratory work of predecessors [36], and based on the HMM, CARM was proposed to identify nine large-scale motion states. Other popular traditional ML techniques include KNN [36,37] and SVM [1,38]. However, ML systems, which are partly dependent on feature engineering and limited by their capability of expression, cannot fully exploit the advantages of big data.

The emergence of DL has helped overcome this bottleneck [39]. Some studies [15,16,17,19] abandoned physics and directly employed CSI for training. LCED [17] uses LSTM to encode preprocessed CSI timing information and then uses a CNN as a decoder to implement a HAR system with user robustness. Moreover, to compensate for cross-domain performance, numerous studies such as EI [40] and CrossSense [41] have attempted to improve the generalization of the model through adversarial learning (AL) or transfer learning (TL). These works rely on super-large-scale data and secondary training (data generation of AL or fine-tuning of TL) to avoid domain noises.

Without this reliance, other studies [12,13] have been conducted in physics. Prior knowledge of spectral features from CSI cooperates with neural networks, which are available for training domain-independent models on small-scale datasets and alleviates unnecessary efforts. Recently, Widar3.0 [13] has become the most influential production method. First, the BVP is derived from Doppler frequency shift profile (DFSP) by using projection, coordinate transformation, and compressed sensing techniques. Subsequently, BVP sequences were modeled by the framework, which combines a gated recurrent units neural network (GRU) with a CNN to construct a novel HGR system (CNN+GRU).

This research focuses on using a Transformer-like backbone to improve the robustness of the HAR system in data-driven methods. We have provided the review directly related to this research in the order of developing progress. However, in this way, some outstanding works in this field will inevitably be neglected. All crucial research is supplemented in Table 1 for comprehensiveness.

### 2.3. Transformer and Attention

Selecting an effective feature extractor is a critical factor in determining the performance of a data-driven method. In this paper, we focus on models that can capture spatial and temporal features, which are determined by the data structure of CSI. While basic machine learning models such as principal component analysis (PCA) [43] are insufficient for complex recognition systems, neural networks can flexibly handle high-dimensional features and can even integrate data from multiple dimensions or modalities. For instance, Autoencoders [44] can automatically reconstruct data in lower-dimensional feature space, while Word2Vec [45] can use contextual information to learn distributed representations of sequence elements and construct the corresponding data features.

CNN and RNN are among the most classical neural networks, which play a leading role in various fields [46,47,48]. The algorithms supported by them are welcomed by the communities of CV and NLP [39] as spatial or temporal feature extractors.The emergence of Transformers [21] and ViTs [49] has diminished the “monopolization” of RNN and CNN. The performance of multi-head self-attention (MSA) creates new opportunities for many research works [50], which is the core of the Transformer and ViT and replaces convolution and recursion.

In fact, attention has been evolving for over 10 years [51]. Research related to WiFi-based perception has begun to attract attention to optimize model performance [15,22,24], not only in the fields of CV and NLP. THAT [22], built a Transformer-like model with a multi-scale convolution based on a two-stream structure. Thus, the time-over-channel features can be used to reinforce the effectiveness and efficiency of the system. However, these efforts either combine attention or Transformers with other networks or replace some components of other backbone networks but keep the structure unchanged.

## 3. Preliminary

### 3.1. From CSI to BVP

CSI is the underlying low-level signal feature of WiFi. Combined with the Doppler effect, CSI enables the inference of the DFSP, a high-level signal feature that reflects signal power variations caused by motion. However, DFSP is dependent on the domains. Using a coordinate transformation, projection, and compressed sensing [52], the domain-independent, high-level, signal feature BVP can be estimated from DFSP [13]. With BVP, domain-independent HAR systems can be realized even without large-scale cross-domain datasets. Figure 2 illustrates the flow from CSI to DFSP and then to BVP.

First, CSI is obtained using the COTS Intel 5300 network interface card (NIC) and the corresponding drivers. This reveals how WiFi signals propagate in wireless channels under the influence of time delay, energy attenuation, and phase shift. A WiFi signal is divided into 30 subcarriers using orthogonal frequency-division multiplexing (OFDM). CSI collected on each subcarrier is a complex-valued channel frequency response (CFR) matrix, where the matrix $H_{i}$ of the *i*-th subcarrier is as follows:(1)$$H_{i}=∥H_{i}∥e^{j∠H_{i}},$$
where $∥H_{i}∥$ and $∠H_{i}$ are, respectively, CSI amplitude and CSI phase. e is a natural logarithm. In theory, the CSI of each subcarrier has orthogonal information corresponding to the same stimulus. Therefore, the CSI triggered by human activity corresponds to a unique signal pattern, as shown in Figure 3. In indoor environments with multiple signal paths, owing to reflection, refraction, and diffraction, CSI is the CFR of the *k* paths at time *t* and frequency frequency *f*.
(2)$$H\left(f,t\right)=e^{j\epsilon (f,t)}\sum_{k=1}^{n}a_{k}\left(f,t\right)e^{-j2\pi f\tau_{k}\left(t\right)},\;n\in N^{*},$$
where $a_{k}\left(f,t\right)$ is an expression of the amplitude fading and initial phase offset of the *k*-th path; $e^{-j2\pi f\tau_{k}\left(t\right)}$ is the phase offset caused by the time delay $\tau_{k}\left(t\right)$ on the *k*-th path; $e^{j\epsilon (f,t)}$ is the phase error caused by factors such as time alignment bias and sampling frequency bias, where *j* is imaginary unit.

The Doppler effect is a natural law. This contributes to the change in the signal propagation distance introduced by the relative movement or obstacle interference between the signal transmitters and receivers. In this process, the Doppler frequency shift (DFS) caused by the reflection of a moving subject can be determined using the following equation:(3)$$f_{D}\left(t\right)=-\frac{1}{\lambda }\frac{d}{dt}d\left(t\right)=-f\frac{d}{dt}\tau \left(t\right)$$
where λ is the wavelength, *f* is the carrier frequency, τ(t) is the time delay, and d(t) is the propagation distance along the non-line-of-sight (NLoS) path. In a multipath environment, considering DFS, CSI can be expressed as the superposition of multiple path responses generated by moving subjects, as follows:(4)$$\hat{H}\left(f,t\right)=\left(H_{s}\left(f\right)+\sum_{k\in P_{d}}a_{k}\left(t\right)e^{j2\pi \int_{-\infty }^{t}f_{D_{k}}\left(u\right)du}\right)\cdot e^{j\epsilon (f,t)},$$
where $H_{s}$ is the sum of all static signals with zero DFS ($f_{D}\left(t\right)=0$), such as line-of-sight (LoS) signals; $P_{d}$ is a set of dynamic signals with non-zero DFS ($f_{D}\left(t\right)\neq 0$); $f_{D_{k}}\left(u\right)$ is the DFS caused by the reflected signal of the moving subject.

Second, by considering Equation (Equation 2), the intrinsic defects of the sampling process must be eliminated. Phase errors $e^{j\epsilon (f,t)}$ are caused by the unmeasurable time and frequency offsets in the CSI. These unknown offsets can be effectively erased by calculating the conjugate multiplication between two antennas on the same WiFi receiver [6]. In addition to the equipment noise, the static component without a dynamic response also contains relatively higher energy. Hence, subtracting the mean of the results of conjugate multiplication from the results is an effective approach to reduce static component noise [10]. Then, the available principal components in the CSI stream can be obtained by PCA because of the high correlation between the high-amplitude pulses and sudden random noise in all subcarriers. Finally, these principal components generate the power distribution spectrums in the time and Doppler frequency domains through a short-time Fourier transform (refer to Figure 4).

The profile of the spectrum is DFSP. In particular, considering HGR as an example, the body and arms move at different speeds when the recognition subject performs gestures. The multipath signals resulting from the gestures perform DFS and are superimposed on the receiver through different links. The spectrum snapshot (profile) generated along the time axis by the spectrum of these signals was defined as DFSP. DFSP is an F×M dimension matrix, where *F* is the frame number of sampling in the frequency domain and *M* is the number of links between transceivers (refer to Figure 5).

Third, combined with DFSP, the domain-independent feature BVP can be inferred [13]. To simplify the reasoning process of BVP, the researchers have transformed the location for calculation from the environmental global coordinate system (ECS) to the body local coordinate system (BCS) (such as the coordinates of the WiFi transceiver) through coordinate transformation. Simultaneously, the BVP matrix $V_{\mathrm{BVP}}$ is quantized as a discrete matrix with N×N, where *N* is the number of potential values of the velocity components decomposed along the axis in the BCS. The range and resolution of the velocity were set as [−2m/a,+2m/s] and 0.2m/s respectively, based on the experimental experience. In the BCS, the velocity component $\vec{v}=\left(v_{x},v_{y}\right)$ is projected onto the frequency component of the DFSP (Figure 5, the origin is the location of human, and the positive direction of the *x* axis is consistent with the human orientation). The projection of the BVP onto the *i*-th link is unique, because the projection coefficient depends only on the location of the *i*-th link (the location of the transceiver in BCS). Based on this, a continuous mapping relationship can be established between DFSP and BVP.

Finally, introducing compressed sensing leads to the estimation of BVP to be a $L_{0}$ optimization problem to obtain the BVP matrix $V_{\mathrm{BVP}}$, which is directly mapped to the gesture motion (as shown in Figure 6).

### 3.2. Multi-Head Self-Attention

Similarity is the basis of self-attention (SA) as a type of extractor for the Transformer-like neural networks. Specifically, the cosine similarity between the current object (vector) and itself and other objects in the input sequence I is output as the similarity score matrix $Q\cdot K^{⊤}$; then the score matrix is applied to the projection V of the original input sequence; at last, the neural network can learn which features need to be focused on through the extent of similarity among the vectors (refer to Equation (Equation 5)).
(5)$$\mathrm{Self}-\mathrm{Attention}\left(Q,K,V\right)=\mathrm{softmax}\left(\frac{Q\cdot K^{⊤}}{\sqrt{d_{k}}}\right)\cdot V.$$

Considering a sequence data I as an example that comprises some vectors, where Q, K, V are three different projections of I that are learnable; $Q\cdot K^{⊤}$ is used to compute the cosine similarity between each vector and others located in other positions as a similarity score matrix; $\sqrt{d_{k}}$ is a scaling factor and the statistic variance, which is used to prevent the failure of softmax in the process of normalization; therefore, the attention score $\mathrm{softmax}(\frac{Q\cdot K^{⊤}}{\sqrt{d_{k}}})$ can be applied to the original projection V of I for training models to learn data features among vectors.

MSA is an optimization of the SA. To enable the models to focus on various messages contained in a group of matrices, MSA decomposes the channel (depth information) of the sequence projection into multiple heads. Multi-head understanding refers to multiple kernels that can generate multiple feature maps in a CNN.

## 4. WiTransformer

### 4.1. System Overview

WiTransformer includes two frameworks: SST and UST. They are designed to model or even learn the features of spatiotemporal signal data. The former uses two Transformer-like encoders for modeling, one for time and the other for space. By contrast, the latter, as the final model, adopts only one encoder with the help of tips. For the expansion of WiTransformer, in this section, the brief of the entire HGR system is essential (as shown in Figure 7). The system consists of four parts: data acquisition, tuning, feature extractor, and classifier.

For data acquisition, a subject surrounded by transceivers (at least three pairs of links) performs gestures in a sensing area. The CSI influenced by the subject, is collected and recorded by the receiver. Subsequently, referring to Section 3.1, the CSI is transformed into BVP in two steps. For Tuning, the generated BVP sequence must be normalized and sequence-aligned (referring to Section 5.1, paragraph “Tuning”). For the feature extractor, SST and UST are leveraged to extract data features that provide recognition (refer to Section 4.2 and Section 4.3). Any classifier is available for the features. We chose the multi-layer perception (MLP, referring to Section 4.5).

Furthermore, inspiration from the feature extractor, which is the core of the system, is interpreted in advance. Considering the character of a BVP sequence (refer to Section 3.1), SST and UST are two reasonable model architectures. In the SST, a BVP data stream is regarded as a video stream for the performance of a gesture. Each frame of the BVP sequence is equivalent to each video image. Accordingly, two encoders are separately required to extract the spatial features per frame and the temporal features of the entire sequence. Subsequently, both features are fused as inputs to the classifier (refer to Section 4.2). In UST, the BVP data stream is regarded as an image whose channels comprise BVPs to ensure that spatial and temporal information can be fused before embedding. Subsequently, spatiotemporal-fused data become the inputs of the extractor (refer to Section 4.3).

### 4.2. Separated Spatiotemporal Model

Figure 8 shows the architecture of SST. The sequence of BVPs is similar to that of a video stream. The temporal and spatial information is modeled in two steps, which draws experience from the structure of long-term recurrent convolutional networks (LRCNs [53]).

The first step was to extract the spatial features from each frame of BVP sequences using spatial encoders. A 2D BVP $V_{\mathrm{bvp}}\in R^{H\times W\times C}$ should be transformed into a 1D vector (token) because the inputs of the standard transformer encoder is a series of tokens embedded in 1D feature space $T_{s}\in R^{N_{s}\times D_{s}}$. Therefore, a frame of BVP is reshaped into a series of flattened 2D patches $x_{p}\in R^{N_{s}\times \left(P^{2}\cdot C\right)}$. H=W=N is the number of possible values of the velocity components in BVP (refer to Section 3.1). $P^{2}$ denotes the patch resolution. $N_{s}$ is the number of patches, and $N_{s}=\frac{H}{P}\cdot \frac{W}{P}={\left(\frac{N}{P}\right)}^{2}$, which represents the available input sequence length for the spatial encoder. *C* is the channel representing the depth information of BVP and C=1. Owing to the latent and invariable vector size $D_{s}$ throughout the transformer encoder, the flattened patches are projected onto $D_{s}$ dimensions using a trainable linear projection $E\in R^{\left(P^{2}\times C\right)\times D_{s}}$ (refer to Figure 9). This process is known as patch embedding.
(6)$$T_{p}^{1};T_{p}^{2};\cdots ;T_{p}^{N_{s}}=x_{p}^{1}E;x_{p}^{2}E;\cdots ;x_{p}^{N_{s}}E.$$

Consequently, each BVP frame was represented by $N_{s}$ tokens. Inspired by BERT’s [class] token [54], [CLS] tokens are preset at the head of every token sequence, for integrating the spatial information of a BVP into one representation vector, Additionally, a learnable encoding $P_{{\mathrm{learnable}}_{1}}$ with $\left(N_{s}+1,D_{s}\right)$ dimensions [55], as the spatial position embedding, are added to the patch embedding to save positional information among patches. Therefore, the inputs of a spatial transformer encoder is as follows:(7)$$y^{t}=\left[T_{s-\mathrm{CLS}}^{0};T_{p}^{1};T_{p}^{2};\cdots ;T_{p}^{N_{s}}\right]+P_{{\mathrm{learnable}}_{1}}\;t=1,\cdots ,T.$$

In the second step, token embedding was used to model the temporal features for the entire 3D BVP sequence $S_{\mathrm{bvp}}\in R^{T\times H\times W\times C}$. Except for the frame number, the *T* is also denoted as the number of [class] tokens from the spatial encoder. Naturally, the spatial [CLS] token sequence can be input into the temporal transformer encoder. Similarly, a temporal [CLS] token is preset at the head of the spatial [CLS] token sequence, and another learnable position encoding $P_{{\mathrm{learnable}}_{2}}\in R^{\left(T+1\right)\times D_{s}}$ is added to the token embedding. Therefore, the inputs to the temporal transformer encoder are as follows:(8)$$y_{0}=\left[T_{t-\mathrm{CLS}}^{0};T_{s-\mathrm{CLS}}^{1};T_{s-\mathrm{CLS}}^{2};\cdots ;T_{s-\mathrm{CLS}}^{T}\right]+P_{{\mathrm{learnable}}_{2}}.$$

### 4.3. United Spatiotemporal Model

SST is an intuitive tactic, but BVPs, such as grayscales, have limited depth information to deduce expressive feature maps. It is difficult for the SST to derive spatially fruitful expression, which will be expanded in Section 5.2. In addition, SST has a quadratic complexity with respect to the number of tokens. This is pertinent to the length of a BVP sequence because the number of tokens increases linearly with the size of the patches and frames. These shortcomings limit the performance of transformer encoders.

Figure 10 shows the architecture of a better choice, UST (in the forthcoming, the meanings of unexplained variables or matrices are consistent with that of Section 4.2). A sequence of BVPs was treated as an image with multiple channels. The image contains not only the spatial features, but also the temporal features in the depth. Therefore, we expanded the traditional two-dimensional image extractor to the spatiotemporal dimension. Then, the spatiotemporal features are packaged into a unified feature vector by stacking. In terms of time sequence, the sequence marking strategy should be taken into account. Finally, temporal and spatial information were fused in a single step, which is enlightened by a 3D CNN [56]. UST is made up of four modules, stacking-fusion, tube embedding, position embedding, and a spatiotemporal encoder (SE).

Stacking-fusion. To strengthen the fitting capability of SST in high dimensional feature space, a 3D BVP sequence $S_{\mathrm{BVP}}\in R^{T\times H\times W}$ is stacked as a 2D “image” $S_{\mathrm{BVP}}\in R^{H\times W\times C}$ whose channels are filled with frames of BVPs (C=T), which fuses the temporal and spatial information in a straightforward manner. However, BVPs must be distinguished from feature maps in digital images. Feature maps do not contain order and interrelationships, whereas BVPs do. For this reason, we selected 3D tubes rather than 2D windows, such as 2D convolution kernels, and proposed channel position embedding (CPE) to compensate for the position information.

Tube embedding. From the perspective of the input of a transformer encoder and the characteristic of the BVP sequence, a BVP “image” is split and reshaped into a sequence of flattened 3D tubes $x_{t}\in R^{N_{t}\times \left(h\cdot w\cdot c\right)}$. (h,w,c) is the size (channel, height, and width) of each tube, and c=C. $N_{t}$ is the number of tubes and $N_{t}=\frac{H}{h}\cdot \frac{W}{w}=\frac{N^{2}}{h\cdot w}$, which serves as the available input sequence length for the SE (refer to Figure 11). Similarly, the flattened tubes were projected onto $D_{t}$ dimensions by using a trainable linear projection $E\in R^{\left(h\times w\times c\right)\times D_{t}}$ (refer to Figure 10):(9)$$T_{t}^{1};T_{t}^{2};\cdots ;T_{t}^{N_{t}}=x_{t}^{1}E;x_{t}^{2}E;\cdots ;x_{t}^{N_{t}}E.$$

Position embedding. The calculations of MSA are based on the pairwise similarity between two elements of a sequence (see Section 3.2), which does not contain any sort or position information. Consequently, we employed frame position embedding in SST. Similar to SST, a group of learnable position encoding $P_{\mathrm{learnable}}\in R^{\left(N_{t}+1\right)\times D_{t}}$ is added to the tube embedding, and a [class] token ∗ was preset. The position-embedded one-dimensional token sequence is the input of the encoder of UST:(10)$$y_{0}=\left[T_{*}^{0};T_{t}^{1};T_{t}^{2};\cdots ;T_{t}^{N_{t}}\right]+P_{\mathrm{learnable}}.$$

However, in stacking-fusion, channel ranking is neglected. The BVP sequence differs from the feature maps of an image without order. Therefore, a group of 3D one-hot matrices $I_{\mathrm{one}-\mathrm{hot}}$ with (h,w,c) dimensions was directly added to the tubes for fewer learnable parameters. This is CPE, a simple and effective tip.
(11)$${x^{\prime }}_{t}^{n}=x_{t}^{n}+I_{\mathrm{one}-\mathrm{hot}}^{n}\;n=1,2,\cdots ,N_{t}.$$

### 4.4. Encoder

The principle of the encoder is that there is no difference between SST and UST. This section elaborates on these points. The transformer encoder is mainly composed of MSA and MLP blocks (refer to Figure 12). Batch normalization (BN) [57] was applied before every block, and residual connections were applied after every block [58]. The MLP block is similar to the MLP layer in the classifier, as described in Section 4.5.
(12)$$y_{\ell }^{\prime }=\mathrm{MSA}\left(\mathrm{BN}\left(y_{\ell -1}\right)\right)+y_{\ell -1},\;\ell =1\ldots L,$$
(13)$$y_{\ell }=\mathrm{MLP}\left(\mathrm{BN}\left(y_{\ell }^{\prime }\right)\right)+y_{\ell }^{\prime },\;\ell =1\ldots L,$$
(14)$$z=\mathrm{BN}\left(y_{L}^{0}\right),$$
where $y_{L}^{0}$ is the non-normalized available input for the classifier and the [class] token ∗. Specifically, the encoder can learn the features of signal power changes in the two-dimensional velocity vector space caused by gesture motion. Multiple heads in MSA enable the network to pay attention to information from different representation subspaces from different time series locations in parallel. In Section 3.1, BVP matrices are stacked along the temporal dimension, placing BVPs of varying temporal positions into equivalent representation subspaces representing the same action. Thus, UST requires only one step to extract spatiotemporal joint features through early fusion.

### 4.5. Classifier

The outputs of the SST and UST are both projected as $N_{g}$ representation vectors through the MLP layer. These representations correspond to gesture categories. The MLP contains two layers with a ReLU non-linear function to optimize the fitting ability. A cross-entropy loss function was used as the loss function. The classifier also includes some vital optimizers involved in model training, which will be introduced in Section 5.1.

## 5. Evaluation and Discussion

This study aimed to probe task complexity changes using pure Transformer-like architectures. Accordingly, we propose two models. In this section, we evaluate these models using experimental results and discuss and analyze the rationality of the correlative design.

### 5.1. Experiments

**Implement.** WiTransformer was developed based on Pytorch [59]. The models are trained and deployed on a small workstation equipped with an NVIDIA Titan RTX graphics card. The details of the hardware implementation scheme are outlined in Widar 3.0, which is not the focus of this study. Overall, at least three pairs of WiFi transceivers were deployed in the sensing area. Each device is equipped with an Intel-5300 NIC. The transceivers were set in monitor mode at 5.825 Hz on channel 165. Only one transmitter antenna was used, which sent 1000 packets per second. The receiver has three antennas that receive signals simultaneously.

**Dataset.** We evaluate WiTransformer on the public dataset Widar 3.0 [60] (called the “original dataset”), which is comprised of 22 gestures from 17 users in 8 locations and 5 orientations in 3 rooms (the data distribution of gesture instances, the schematic diagram of all gestures, and labels are shown in Figure 13 and Figure A1, and Table 2). The resulting data imbalance is shown in Figure 13. The experiment is not significantly affected unless the sample ratio between the two categories is at least 1:10.

The original dataset was reconstructed as four task datasets (TDSs) with qualitatively increasing recognition complexity to test the recognition accuracy and robustness of the models with changing task complexity. The basics of reconstruction are presented in Table 3. Specifically, the higher the **Classification** number, the lower the distinguishing ability [61,62]; assessing the process of execution, the higher the group number of gestures with similar (**Similarity**) motions (e.g., gesture motion “Draw-2” and gesture motion “Draw-Z”), the lower the distinguishing ability; and the higher the number of gesture with **Distortion** posed by the lack of vertical resolution for BVP, the lower the distinguishing ability [13].

The reconstructive scheme is presented in Table 4. Specifically, **TDSs-6** has 6 non-confusing gestures; **TDSs-9** has 4 groups (separated by semicolons in Table 3) with similar gestures and a higher classification number compared with TDSs-6; **TDSs-12** has 12 gestures, including 2 groups of similar gestures and 3 vertical gestures with distortion; and **TDSs-22** has 22 gestures with semantic information and all task complexities compared with the above task datasets.

**Metrics.** The recognition accuracy (Acc) and F−1 scores are comprehensive indices measuring the recognition performance of the models (as shown in function (Equation 15) and (Equation 16)):(15)$$\mathrm{Acc}=\frac{\;\mathrm{TruePositive}+\mathrm{TrueNegative}}{\;\mathrm{TruePositive}\;+\;\mathrm{FalsePositive}\;+\;\mathrm{TrueNegative}\;+\;\mathrm{FalseNegative}\;},$$
(16)$$F-1=\frac{2\times \mathrm{Prc}\times \mathrm{Rec}}{\mathrm{Prc}+\mathrm{Rec}}.$$
(17)$$\mathrm{Prc}=\frac{\;\mathrm{TruePositive}\;}{\;\mathrm{TruePositive}\;+\;\mathrm{FalsePositive}\;},$$
(18)$$\mathrm{Rec}=\frac{\;\mathrm{TruePositive}\;}{\;\mathrm{TruePositive}\;+\;\mathrm{FalseNegative}\;},$$
where Prc is the precision, Rec is the recall. To remedy errors due to training settings, such as parameter initialization, the k-average accuracy is calculated as the final result by *K* times shuffling testing data, $\bar{\mathrm{Acc}}=\frac{1}{K}\cdot \sum_{k=0}^{K}{\mathrm{Acc}}_{k}$ [17]. Considering the slight data imbalance, the weighted F−1 score can be considered as a supplement, which is the weighted average of the F−1 scores for all categories according to the sample number of each category. Furthermore, the standard deviation (SD) of the sample volume was used as a quantitative metric of data imbalance, where the higher the SD, the more unbalanced the data (refer to Table 3). Except for the typical metrics, ${\mathrm{Rate}}_{\mathrm{drop}}$, the accuracy dropout rate was used as an indicator to evaluate the robustness of resisting the task complexity change.

**Tuning.** According Widar 3.0, two barriers must to be handled in practical experiments to simplify the learning model. First, the total power of BVP, as a type of power spectrum, may change, which is affected by the inherent transmission power adjustment mechanism of WiFi for communication. Second, the time and speed of gesture execution can hardly be quantified accurately. Gestures from different subjects (even the same subject) may have discrepancies, such as inconsistent power ranges and misaligned sample sequences, which are attributed to the above barriers.

For signal power variation, WiTransformer followed the 2D matrix normalization proposed by Widar3.0. All the BVP matrices are processed separately along the time sequence such that the sum of all power values in each frame of the BVP sequence is one. For the execution time discrepancy, we performed an optimization with padding, compared with Widar 3.0, which requires some assumptions and additional computation. All BVP sequences are padded with zero matrices to the same length, $t_{\max}$ being the longest sequence inspired by machine translation.

**Training.** In each TDSs, 80% of the data were used for training each type of gesture. The basic training settings were as follows. The training data were randomly selected in batches (the batch size was set to 64) from the training set as the input. A convergence threshold is set for the loss function to overcome over fitting. Specifically, if the reduction in the loss value is below $10^{-2}$ during 20 consecutive training epochs and the training accuracy is over 95%, the training will be reduced. Adam is the preferred choice for optimizing tasks. For efficient convergence, we used a multi-step learning-rate scheduler, $Lr_{e}=Lr_{e-1}\cdot \gamma^{s(e,m)}$, to adjust the learning rate, where $e=1,\cdots ,N^{*}$ is the number of training epoch; $m\in N^{*}$ is the total step length; γ is the scaling hyperparameter; and $s\left(e,m\right)=\frac{m}{e}$ is a sequence related to *e* and *m* (refer to Figure 14, the adjusting hyperparameter γ is set with 0.9). $L_{2}$ regularization and dropout are introduced to balance the complexity of the model for generalization. The regularization coefficient was set to 0.001. A dropout is applied to the output of each sub-module (embeddings, MSA, MLP, and encoder). The initial model weight is initialized using the Kaiming initialization [63].

In addition to the parameters above, the model framework and hyperparameters (for the final model UST) directly related to the model structure are listed in Table 5. The ablation of these hyperparameters is described in Section 5.5.

### 5.2. Effectiveness

ViT showed that Transformers lack the inductive biases inherent to CNNs and therefore do not generalize well when trained on small datasets [49]. In fact, the data scale of Widar 3.0 (approximately 500–3500 pieces of data per category) is smaller than the datasets mentioned by ViT (ImageNet-21k [62] and JFT [64], with over 700 and 17,000 pieces of data per category, respectively). Transformer-like models are unavailable for the Widar 3.0 dataset.

Meanwhile, in Section 4.3, we predicted that SST, as a Transformer-like model lacking inductive biases owing to the limited depth information, could hardly fit fruitful features. Furthermore, the two separate transformer encoders of the SST magnify this flaw.

These two points make the effectiveness of applying Transformer-like models to WiFi-based cross-domain HAR systems ambiguous. However, in Section 1 and Section 3.1, we highlighted that BVP, as a generalized approach from the perspective of physics rather than modeling, overcomes the domain dependence of CSI and DFS. This approach provided a practical basis for these systems. The UST design expands depth information for a transformer encoder, utilizing fewer computing resources than its predecessor, SST.

The discussions in this section highlight the effectiveness of UST and BVP both theoretically and empirically. The tendencies of the loss function and accuracy when the SST and UST were trained on TDSs-22 are shown in Figure 15. This illustrates the results of our analysis. BVPs cannot effectively drive the SST to upgrade the gradient; therefore, the loss quickly reaches a bottleneck. By contrast, UST fuses spatiotemporal information by stacking BVP sequence for more depth information and obtains better convergence (this phenomenon will be confirmed again in Section 5.4). Therefore, UST was regarded as the final proposed model. The recognition results of the classification confusion matrix of UST on TDSs-22 are displayed in Figure 16.

**Remark 1.** 
*In this study, we attempt to use the self-attention of Transformers to counter changes in task complexity. We explored two spatiotemporal encoder models: UST and SST. Based on the discussion in Section 5.2, we can predict that the SST model may fail, although there is no large-scale verification with only one round of training attempts. This is because of the lack of inductive bias in transformer encoders and the difficulty in obtaining large-scale data using WiFi-sensing devices (refer to Section 1, Introduction). Even if we adopt the cross-domain signal feature BVP to reduce the data scale requirements of Transformers, SST contains two sets of transformer encoders, which undoubtedly superimpose the defects of Transformers.*


### 5.3. Model Comparison

In this section, WiTransformer as a backbone network will be compared with popular backbone models, “CNN+LSTM”, “CNN+GRU”, “Conv3D”, and “Conv2D”, in terms of recognition accuracy and robustness, complexity, and efficiency. “CNN+LSTM” and “CNN+GRU” are both separated spatiotemporal models, which are typical backbone networks for HAR [13,17,53]; “Conv3D”, is a 3D CNN is united spatiotemporal backbone network that is fashionable in HAR [65,66], which is not yet applied to WiFi-based HAR; “Conv2D”, a 2D CNN, is the most popular DL backbone.

#### 5.3.1. Accuracy and Robustness

The 10-average accuracy was used as the final result (Figure 17). All models have experienced varying degrees of decreased accuracy with the rise in Classification number, Similarity, and Distortion. Specifically, Conv3D achieved the best accuracies of 90.75% and 89.2% on TDSs-6 and TDSs-9, which provides evidence for the combination of united spatiotemporal architectures and BVPs. Inductive biases make it more generalizable in the verification set. However, this advantage disappears when the task complexity increases because of the lack of global features. UST outperforms Conv3D on TDSs-12 and TDSs-22 (86.86% and 86.16%, respectively), and all the other models. UST can distinguish fine-grained spatiotemporal differences by globally capturing redundant information and long-range dependencies to maintain a relatively stable performance. As previously interpreted, SST is unsuitable for BVPs; therefore, its performance is poor. There was a suppression of TDSs-9 for UST and SST. The lack of inductive biases may make the Transformer-like models sensitive to data imbalances (refer to Table 3). Similar to SST, CNN+LSTM and CNN+GRU as separate spatiotemporal architectures are also unsuitable for BVPs on more complex tasks. As predicted, although Conv2D is a 2D model, it can realize 3D feature extraction with the “stacking-fusion” we proposed in Section 4.3, and the accuracy of the modified “Conv2D” can be comparable to that of “Conv3D”. The effectiveness of early fusion in UST was confirmed again (refer to Section 5.2). Considering the data imbalance, the performance comparison reflected by the weighted F-1 score is consistent with the reflected accuracy (as shown in Table 6).

The superiority of UST cannot be seen directly from Figure 17, which is not capable of accommodating the complexity of the task. ${\mathrm{Rate}}_{\mathrm{drop}}$ represents the robustness against the task complexity change. As shown in Table 7, as the task complexity increases from TDSs-6 to TDSs-22, ${\mathrm{Rate}}_{\mathrm{drop}}$ decreases by approximately 15% to 20% for CNN+GRU and CNN+LSTM, and by approximately 7% for CNNs. It is clear that the ${\mathrm{Rate}}_{\mathrm{drop}}$ of UST decreased by only 3.18%. Its robustness surpasses that of other backbone networks.

**Remark 2.** 
*The results of the comparison and test experiments confirm our previous inference. The performance of SST remains highly dependent on the data scale, but UST can achieve satisfactory performance using BVP. Simultaneously, we believe that owing to data imbalance, SST cannot learn effective bias from instances with fewer samples, thereby significantly reducing the accuracy of TDSs-9. This is verified again by the F-1 score. After iterations, the model habitually maintained a high recognition accuracy through the recall, but precision was always zero.*


#### 5.3.2. Efficiency and Complexity

HAR is an application layer of intelligent sensor network technology. Execution and computing costs determine whether a model can be implemented in practice. This section discusses the paradox between efficiency and efficacy and the model complexity. Deep learning models are typically trained on GPU (NVIDIA TITAN RTX) and deployed on CPU (Intel(R) Xeon(R) Gold 5118 CPU @ 2.30 GHz). Therefore, six backbone models were tested using the two devices to compare their spatial and temporal costs.

Table 8 shows that, for a model, the required scale of floating-point operations (FLOPs) and trainable parameters (Parameters) are trained once in GPU, as well as the inference time (Time) and memory cost (Memory) deployed on the CPU. According to the statement of the original dataset, the gesture execution time was 1.5 s, which was accessible for all models. However, LRCN-like models (“CNN+LSTM” and “CNN+GRU”) are limited by serial units and may not be real-time in actual environments, which is also a potential problem for Widar 3.0. Compared with SST and Conv3D, because UST only uses a one-dimensional encoder and integrates spatio-temporal features before coding, the computational complexity of feature extraction is significantly reduced, and the computing time cost of the CPU is shortened. Conv2D also integrates spatio-temporal features in advance using a stacking-fusion module to reduce computational complexity. In terms of spatial expenses, even Conv3D, which consumes the most memory, can be easily deployed on small mobile devices. For training, SST and Conv3D required more time for larger computing complexities and parameter scale.

In terms of model complexity, the “Big O” notation is used to perform the analysis. The model complexities of the core computing modules for the six backbones are listed in Table 9, where *n* is the sequence length, *d* is the feature representation dimension, and *k* is the kernel or patch size. The complexity of UST and SST mainly comes from pairwise computing between tokens in self-attention: the longer the input sequence, the larger the number of tokens, and the higher the complexity. Nonetheless, the spatial tube sequence size (*n*) of the split BVP matrix is fixed such that the size *n* is constant for UST but not for SST. When the sequence length *n* is less than the numerical size *k* of the convolution kernel, the complexity of UST is lower than that of Conv2D. The backbones utilizing recurrent units as the calculation core were constrained by the sequence length. The longer the execution time of the gesture instance, the more complex the model was.

In a comprehensive evaluation of accuracy, task robustness, and execution efficiency, the UST of WiTransformer emerges as an appropriate model with relatively minimal training and deployment consumption, along with high recognition performance. On small-scale datasets, such as TDSs-6, it achieves similar or better results than existing models and state-of-the-art methods. As the task complexity of the dataset increases, it maintains the lowest performance loss and demonstrates the best task robustness. These achievements do not come at the cost of increased development expenses compared to other models, and it still maintains high computational efficiency.

### 5.4. Comparison with State-of-the-Art

In this section, we compare the final model, UST, with several representative state-of-the-art models, CARM [6], Widar3.0 [13], LCED [17], and THAT [22] on TDSs-6. Despite the discrepancy in hyperparameter settings and inputs, a fair comparison cannot be drawn. However, this can still justify the advancement of WiTransformer in WiFi-based HAR.

Specifically, CARM uses DFS as the input to train the HMM. Widar3.0 trains an LRCN-like model combining CNN with GRU by using BVP. LCED uses LSTM as an encoder in time and CNN as a decoder in space features with the filtered CSI data used directly for training. THAT uses two transformer encoders to build a two-stream framework for extracting features from raw CSI in the time and frequency domains.

Twenty recognition results for the validation sets are showed in Figure 18. The UST of WiTransformer and Widar 3.0 achieve equivalent performance on TDSs-6 by relying on the cross-domain feature BVP. Owing to weak domain independence, the recognition performances of CARM, LCED, and THAT were inferior on the cross-multi-domain dataset. However, CARM lags behind the others because of its being short feature-expression ability.

### 5.5. Ablation

In this section, hyperparameter ablation and module ablation are presented to ablate the redundant structure of UST and get the best hyperparameter configuration, hyperparameter ablation, and module ablation, respectively.

#### 5.5.1. Module Ablation

A module ablation study was conducted on TDSs-6 to assess the contributions of optional modules closely related to performance such as dropouts, normalizations, CPE, and position encodings (PE). For each ablation, a specific component is ablated from the entire model. The results show that all these modules contributed significantly to the model performance (refer to Table 10).

We conducted additional ablations for the generalization of CEP. We used Conv2D and UST, respectively, in two modes with and without CPE, to test the performance. As shown in Table 11, both networks with CPE have similar boosts. Therefore, the CPE we proposed is independently effective.

#### 5.5.2. Hyperparameter Ablation

Mining empirical parameters that can be modified to improve performance often requires intensive computation to determine the correct network hyperparameters. Hyperparameter ablation tests help us determine which practical settings truly match. We performed ablations by scaling different UST architecture dimensions to determine the best hyperparameter configuration using SMAC [67]. All the configurations are based on the training settings described in Section 5.1. However, manual searching for the parameter space is exclusive and tedious. Thus, we used Microsoft NNI [68] to adjust hyperparameter combinations (refer to Figure 19). The ablated configuration, which is the result of hyperparameter ablation, can be seen in Table 5.

Additionally, we utilized an optimization strategy to eliminate the parametric uncertainty of complex deep models, and applied it to all comparison models. Table 12 lists the parameters of all comparison backbones, where the optimal parameter configuration after the parametric uncertainty ablation is marked with underline and bold. The listed parameters include: “LR_init” for initial learning rate, “weight decay” for the coefficient of $L_{2}$ regularization, “Gamma” for the optimizer parameter Adam, “kernel size”, “stride”, and “dimension-1” for the size of convolutional kernel, stride, and hidden feature dimensions, “maxpooling” for the max-pooling layer in the convolutional block, “dimension-1” and “dimension-2” for the hidden dimensions of full connection layer, “Blk_num-1” and “Blk_num-1” for the number of convolutional and recurrent modules, respectively, “drop_ratio-1” and “drop_ratio-2” for the ratio of dropout layer, and “Act.Fun.” for the activation function for all hidden nonlinear nodes.

## 6. Conclusions

In this paper, we proposed WiTransformers, two transformer-like models, to resist task complexity changes in WiFi-based HAR systems. Unlike previous studies based on neural networks, we eliminate the common convolutional and recursive backbones. Specifically, we explored two spatiotemporal feature extraction methods—SST and UST—using a modified transformer encoder. For UST, we proposed tube embedding and CPE to capture three-dimensional features using a one-dimensional extractor. We then introduced four task datasets reconstructed from the public dataset, Widar 3.0, to evaluate the performance of WiTransformers. Contrastive and validation experiments were conducted on task datasets using CNN+LSTM, CNN+GRU, Conv2D, and Conv3D as comparative objects. The experimental results demonstrate that UST outperforms the other backbones, with accuracies of 86.86% and 86.16% on TDSs-12 and TDSs-22, respectively. Moreover, with an increase in recognition task complexity, the recognition accuracy of the other models decreased by approximately 7–21%, whereas UST only reduced by 3.18%. Finally, we compared UST, as the final model, with related works, such as Widar 3.0, CARM, LCED, and THAT, to present the advancement of UST. The comparison suggests that the 20 times recognition results of UST on TDSs-6 are overall better than those of the state-of-the-art work, Widar 3.0.

In addition to its focus on robust behavior recognition based on WiFi, this research provides valuable insights for future studies in the design of WiTransformer, particularly with regard to its UST architecture. Firstly, as a practical multi-modality fusion tool, the Transformer can be employed to integrate WiFi signals with other modalities such as natural language and digital images for behavior recognition based on multi-modality. Secondly, the approach of stacking-fusion and tube embedding utilized in WiTransformer can be applied to other dynamic recognition tasks, such as activity recognition based on videos and remote sensing.

In future research, we aim to continue to learn, determine, and design better cross-domain cognition methods independent of BVPs. Furthermore, credible and efficient large-scale datasets are also in our work plan.

## Figures and Tables

**Figure 1 sensors-23-02612-f001:**
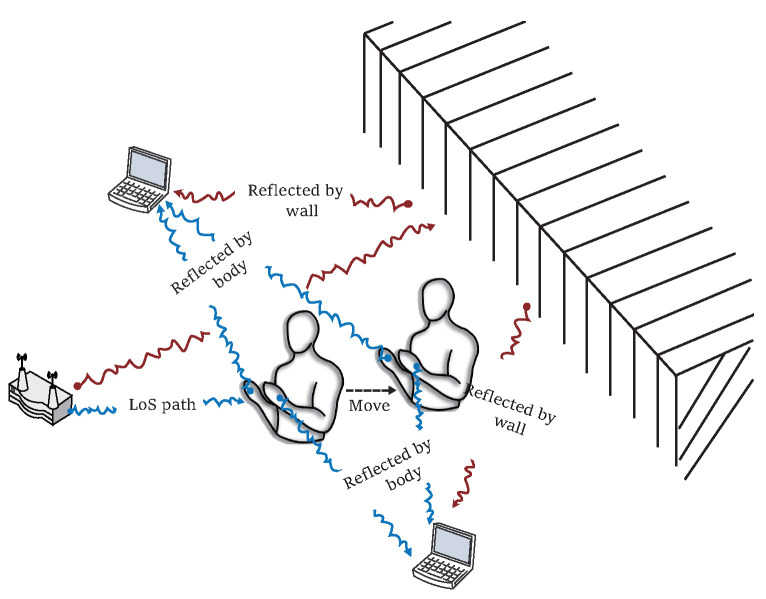
A WiFi signal propagates through multipath in reflected, refracted, and diffracted manners. For brief and clarity, only one way, the reflection, is shown in the illustration, where the red waves represent signals affected by moving torso and the blue waves represent signals affected by static wall.

**Figure 2 sensors-23-02612-f002:**
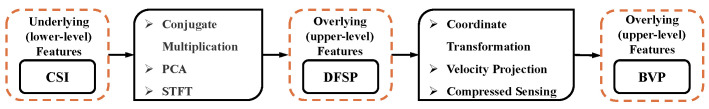
Inference from CSI to DFSP and BVP.

**Figure 3 sensors-23-02612-f003:**
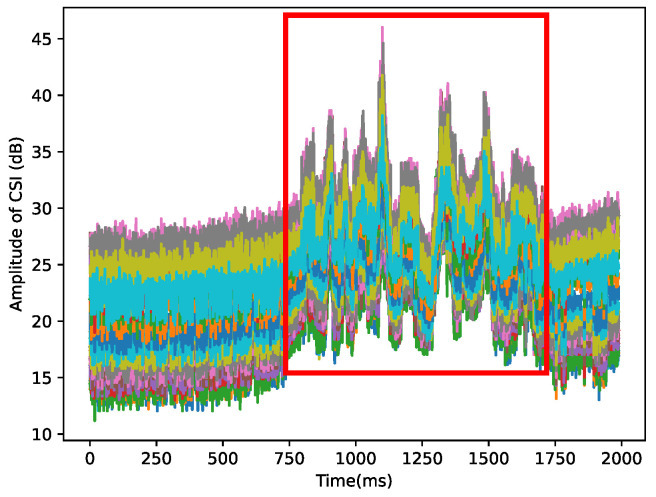
The amplitude of 30 subcarriers are drawn with different colors. These signal patterns, circled by a red box, correspond to the human behavior that causes them.

**Figure 4 sensors-23-02612-f004:**
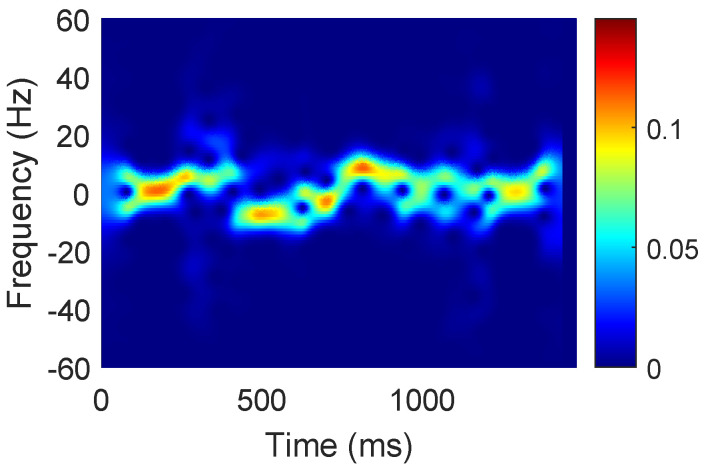
Power distribution spectrum caused by human gestures.

**Figure 5 sensors-23-02612-f005:**
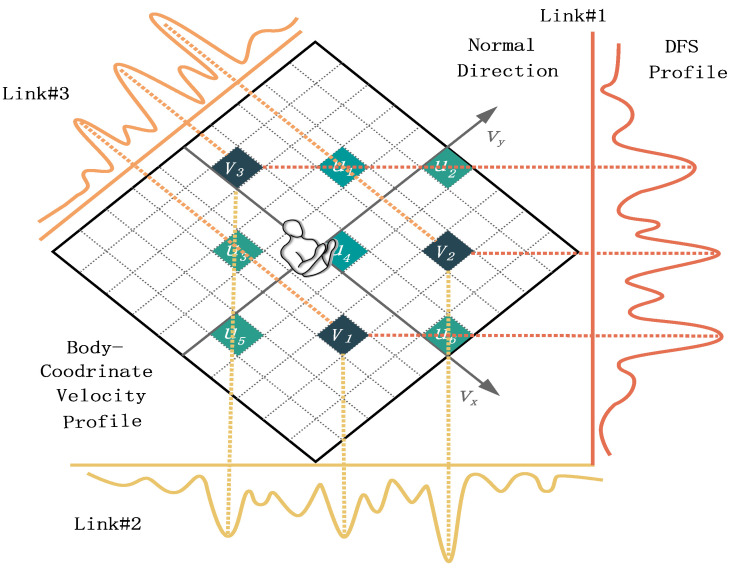
The velocity components of human torso profiles are projected on frequency components to realize the mapping between velocity and signal power, which is the relationship between BVP and DFSP.

**Figure 6 sensors-23-02612-f006:**
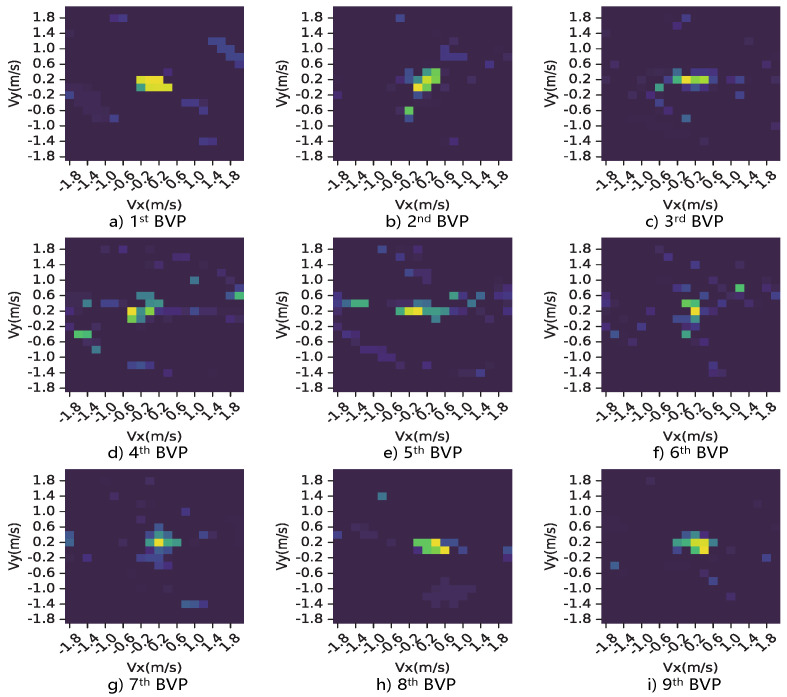
This is a part of BVP sequence corresponding to the motion, “push and pull”, where the principal signal power components are rendered in all snapshots.

**Figure 7 sensors-23-02612-f007:**
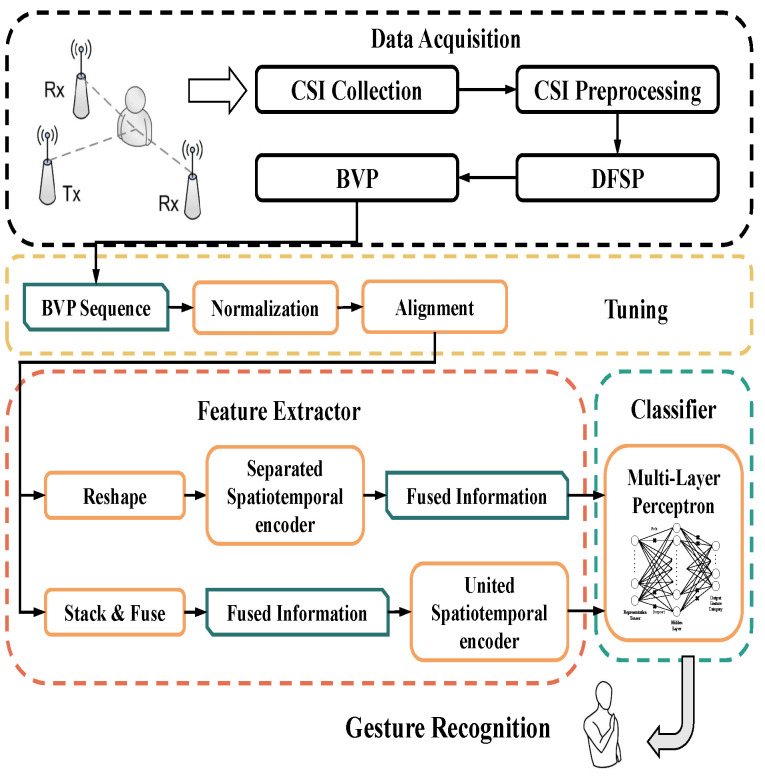
Overall flow chart of the WiTransformer system.

**Figure 8 sensors-23-02612-f008:**
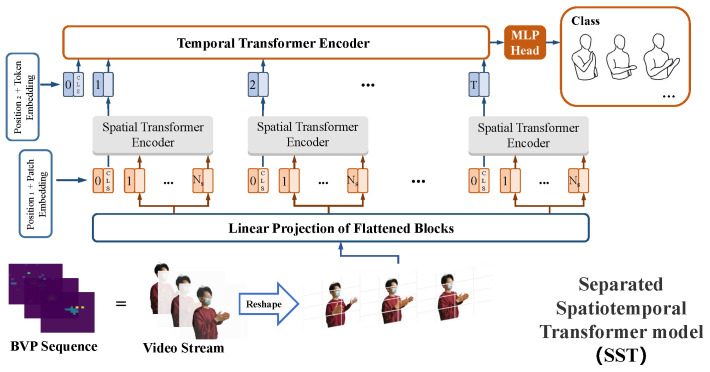
Separated spatiotemporal transformer model.

**Figure 9 sensors-23-02612-f009:**
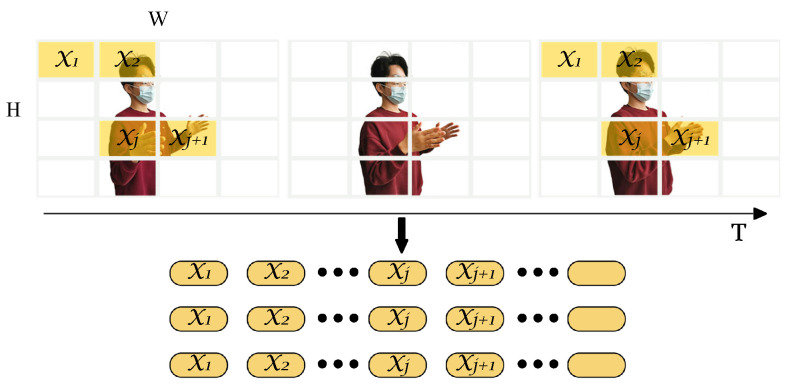
Patch embedding.

**Figure 10 sensors-23-02612-f010:**
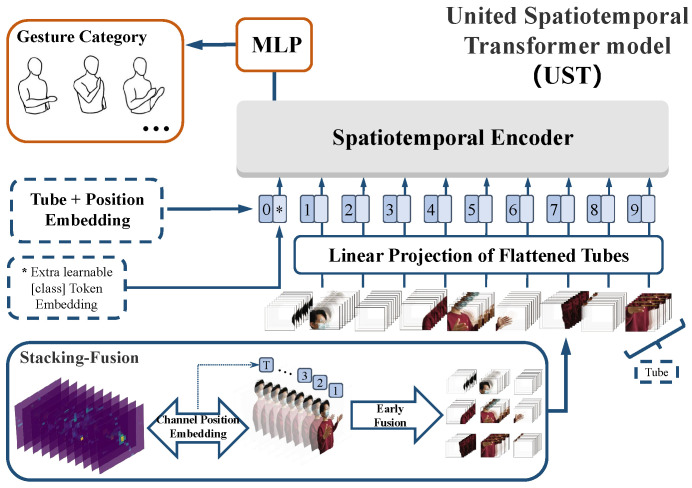
United spatiotemporal transformer model, where * is the [class] token.

**Figure 11 sensors-23-02612-f011:**
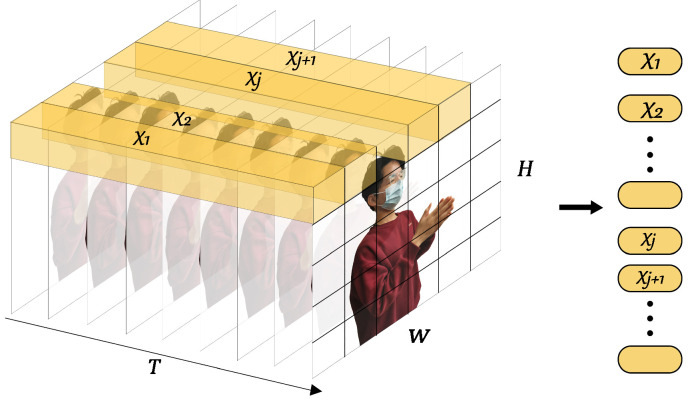
Tube embedding.

**Figure 12 sensors-23-02612-f012:**
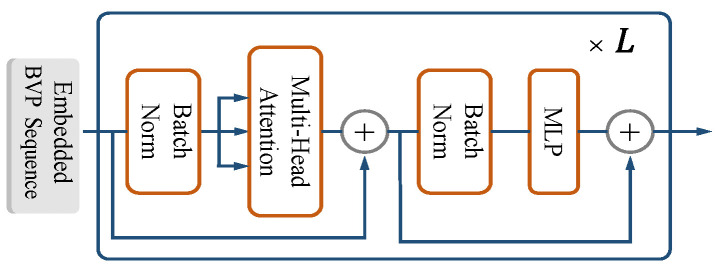
The encoder.

**Figure 13 sensors-23-02612-f013:**
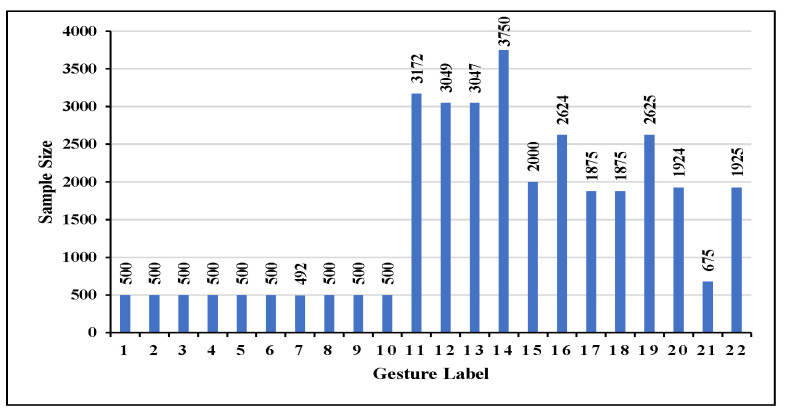
Data distribution of the 22 gestures.

**Figure 14 sensors-23-02612-f014:**
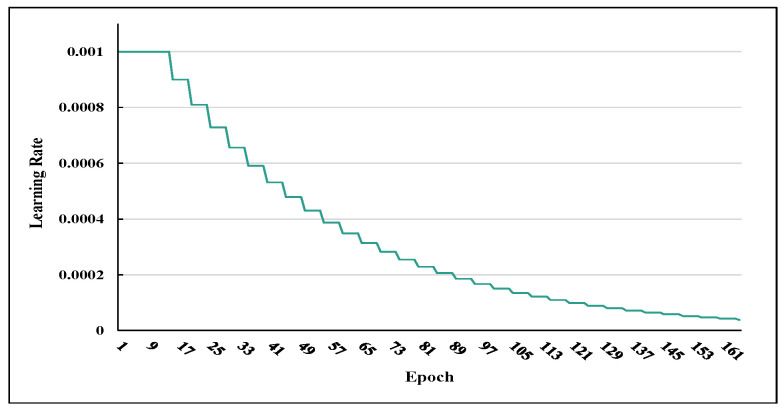
Multi-step learning-rate scheduler.

**Figure 15 sensors-23-02612-f015:**
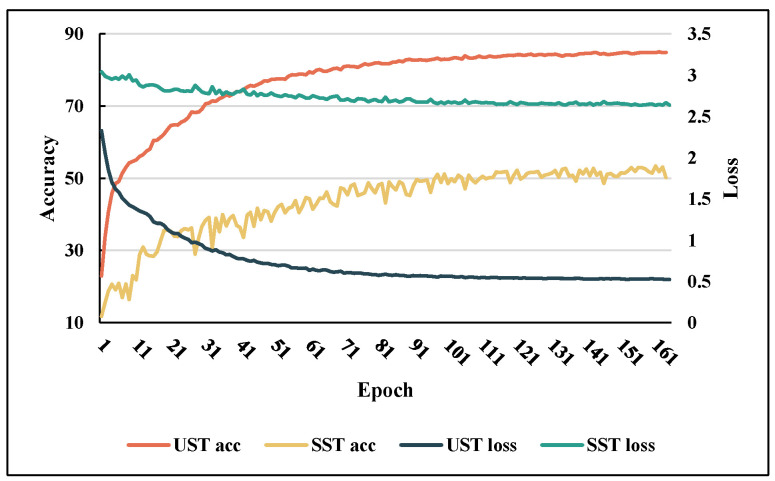
On TDSs-22, the tendencies of the losses and the accuracies of UST and SST.

**Figure 16 sensors-23-02612-f016:**
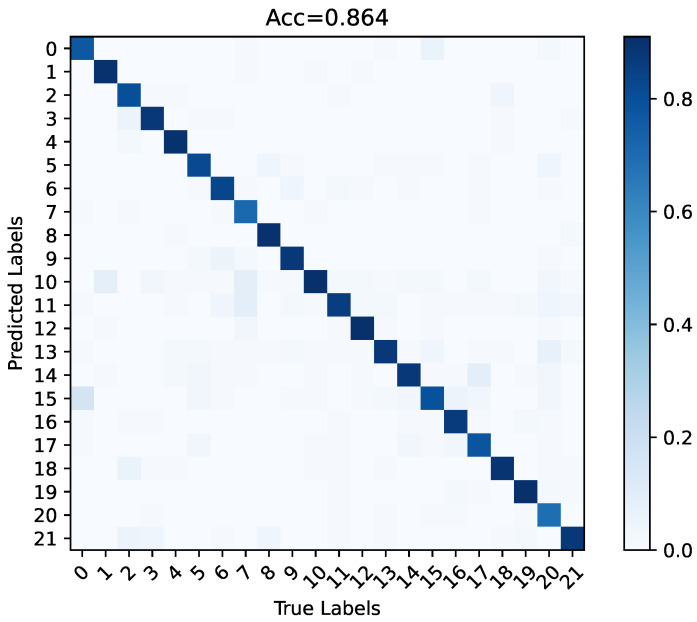
On TDSs-22, the classification confusion matrix of UST.

**Figure 17 sensors-23-02612-f017:**
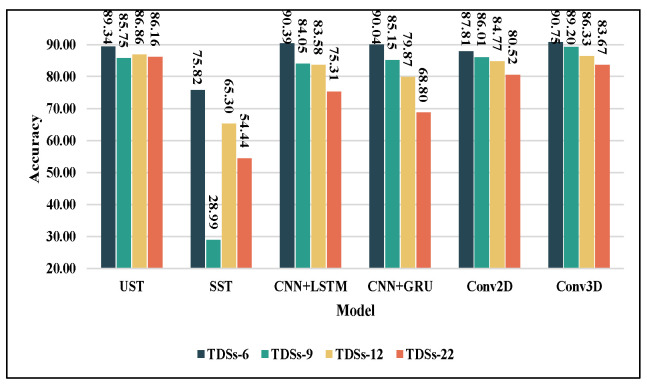
Recognition accuracies and their variations of six backbones on four task datasets.

**Figure 18 sensors-23-02612-f018:**
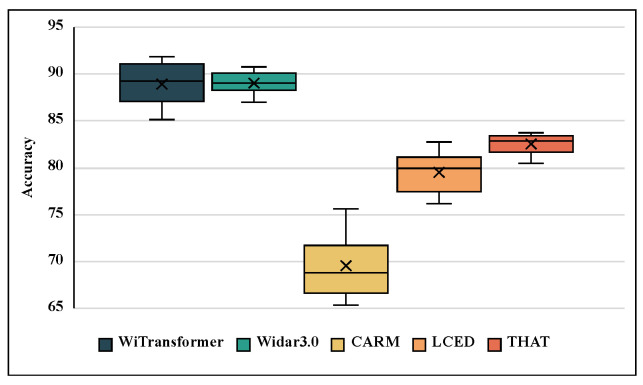
Recognition accuracies of the five methods on TDSs-6.

**Figure 19 sensors-23-02612-f019:**
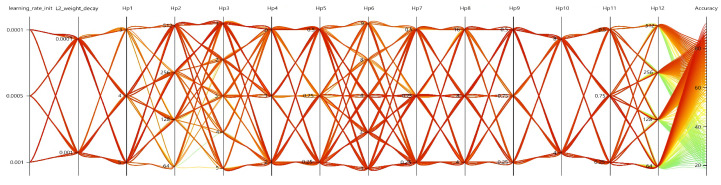
Hyperparameter ablation.

**Table 1 sensors-23-02612-t001:** A survey of the existing deep learning approaches for WiFi sensing.

Method	Task	Model	DataType
EI [40]	HGR	Adversarial Generative networks	CSI
CrossSense [41]	HGR	Roaming model (Transfer Learning)	CSI
Ref. [15]	HAR	BLSTM	CSI
WIHF [12]	Human Identification Gesture Recognition	CNN+GRU	DFS
Widar 3.0 [13]	HGR	CNN+GRU	BVP
LCED [17]	HAR	CNN+GRU	CSI
THAT [22]	HAR	Transformers (Two-stream)	CSI
WiGr [19]	HGR	Prototypical Networks	CSI
Ref. [24]	HGR	Transformers	CSI
WiGRUNT [26]	HGR	CNN (Attention)	CSI
SenseFi [42]	HAR, HGR	ResNet, LSTM Transformers	CSI, BVP
STC-NLSTMNet [20]	HAR	CNN+NLSTM	CSI
AFEE [18]	HAR	MatNet	CSI

**Table 2 sensors-23-02612-t002:** The numerical labels corresponds to the 22 gestures, “H” represents horizontal and “V” represents vertical.

Gesture	Label	Gesture	Label	Gesture	Label
Draw-0	0	Draw-8	8	Draw-Rectangle (H)	16
Draw-1	1	Draw-9	9	Draw-Triangle (H)	17
Draw-2	2	Push and Pull	10	Draw-Z (H)	18
Draw-3	3	Sweep	11	Draw-N (V)	19
Draw-4	4	Clap	12	Draw-O (V)	20
Draw-5	5	Slide	13	Draw-Z (V)	21
Draw-6	6	Draw-N (H)	14		
Draw-7	7	Draw-O (H)	15		

**Table 3 sensors-23-02612-t003:** The reconstructive basic of four task datasets depends on Classification, Similarity, and Distortion. Additionally, the data imbalance is presented here.

TDSs	Classification	Similarity	Distortion	Data Imbalance
TDSs-6	6	0	0	380.09
TDSs-9	9	4	0	1223.80
TDSs-12	12	2	3	790.18
TDSs-22	22	4	3	1102.79

**Table 4 sensors-23-02612-t004:** Four task datasets and the corresponding gesture types.

Task Dataset	Gesture
TDSs-6	Push and Pull, Sweep, Clap, Slide, Draw-O (H), Draw-Z (H)
TDSs-9	Draw-0, Draw-O; Draw-1, Slide, Sweep; Draw-2, Draw-Z (H); Draw-4, Draw-Triangle (H)
TDSs-12	Push and Pull, Sweep, Clap, Slide, Draw-O (H), Draw-Z (H), Draw-N (H), Draw-Rectangle (H), Draw-Triangle (H), Draw-N(V), Draw-O (V), Draw-Z (V)
TDSs-22	Draw-0, Draw-1, Draw-2, Draw-3, Draw-4, Draw-5, Draw-6, Draw-7, Draw-8, Draw-9, Push and Pull, Sweep, Clap, Slide, Draw-O (H), Draw-Z (H), Draw-N (H), Draw-O (V),Draw-Z (V), Draw-N (V)

**Table 5 sensors-23-02612-t005:** Structure, hyperparameters, and their value and hyperparameter labels of UST.

Module	Hyperparameter	Value	Label
Stacking-Fusion	Tube size	*h* = 2, *w* = 2, *t* = $t_{\max}$	Hp1
	Tube Embedding Dimension (*D*)	256	Hp2
	Tube Stride	2	Hp3
	Tube Padding	0	Hp4
	Dropout Ratio 1	0.75	Hp5
Spatiotemporal Encoder	Spatiotemporal Layer Number	1	Hp6
	Droupout Ratio 2	0.25	Hp7
	MSA Head Number	16	Hp8
	MSA Dropout Ratio	0.25	Hp9
	MLP Hidden Dimension	4 × 256	Hp10
	MLP Output Dimension	256	Hp2
	Dropout Ratio 3	0.25	Hp11
MLP Classifier	MLP Projection Dimension ($N_{g}$)	16	Hp12

**Table 6 sensors-23-02612-t006:** F-1 scores of six backbones on four task-sets.

Model Type	TDSs-6	TDSs-9	TDSs-12	TDSs-22
UST	0.8731	0.8538	0.8501	0.8531
SST	0.7314	0.0	0.4122	0.5128
CNN+GRU	0.8633	0.7940	0.6034	07319
CNN+LSTM	0.8726	0.7950	0.6094	0.7147
Conv2D	0.8701	0.8496	0.8165	0.8227
Conv3D	0.8947	0.8640	0.8353	0.8411

**Table 7 sensors-23-02612-t007:** ${\mathrm{Rate}}_{\mathrm{drop}}$s of six backbones on four task-sets.

Model Type	${\mathrm{Rate}}_{\mathrm{drop}}$
UST	−3.18%
SST	−21.42%
CNN+GRU	−21.24%
CNN+LSTM	−15.08%
Conv2D	−7.29%
Conv3D	−7.08%

**Table 8 sensors-23-02612-t008:** Temporal and spatial consumption of the six backbones.

Model	FLOPs	Parameters	Time (CPU)	Memory (CPU)
UST	56.37 M	0.69 M	11.77 ms	3.80 Mb
SST	105.65 M	3.29 M	55.93 ms	16.61 Mb
CNN+GRU	34.96 M	0.84 M	991.88 ms	3.25 Mb
CNN+LSTM	37.73 M	0.94 M	981.25 ms	3.63 Mb
Conv2D	12 M	0.58 M	7.10 ms	6.21 Mb
Conv3D	25.13 M	12.85 M	21.70 ms	49.06 Mb

**Table 9 sensors-23-02612-t009:** Model complexity of the core computing module for the six backbones.

Computing Module	Model Complexity	Sequential Operations	Backbone
Self-Attention	$O(n^{2}\cdot d)$	O(1)	UST
Self-Attention × 2	$O(k^{2}\cdot n^{2}\cdot d)$	O(1)	SST
Recurrent	$O(n\cdot d^{2})$	O(n)	-
2D Convolutional	$O(k\cdot n\cdot d^{2})$	O(1)	Conv2D
3D Convolutional	$O(k\cdot n\cdot d^{3})$	O(1)	Conv3D
CRNN	$O(k\cdot n^{2}\cdot d^{2})$	O(n)	CNN+LSTM, CNN+GRU

**Table 10 sensors-23-02612-t010:** Ablation analysis of the model structure.

Module	Accuracy	Delta (Δ)
UST	89.34%	-
- Dropout	66.58%	−22.76%
- BN	16.34%	−73%
- Learnable PE	68.61%	−20.73%
- CPE	85.58%	−3.76%

**Table 11 sensors-23-02612-t011:** A ablation study regarding the effectiveness of CPE.

Model	CPE	No CPE
UST	89.34%	85.58%
Conv2D	87.81%	84.96%

**Table 12 sensors-23-02612-t012:** The ablation for parametric uncertainty, where the optimal parameter configuration after the parametric uncertainty ablation is marked with underline and bold.

Hyperparameters	CNN+LSTM	CNN+GRU	Conv3D	Conv2D
LR_init	[0.0001, 0.00002, **0.00001]**	[0.0001, **0.00002**, 0.00001]	[0.005, **0.001**, 0.0002]	[0.0002, **0.00005**, 0.00001]
weight decay	[0.001, 0.0001, **0.00005**]	[0.001, 0.0001, **0.00005**]	[0.001, 0.0001, **0.00005**]	[0.001, 0.0001, **0.00005**]
Gamma	[0.9 **0.7** 0.5]	[0.9 **0.7** 0.5]	[**0.9** 0.7 0.5]	[0.9 0.7 **0.5**]
kernel size	[**5**, 3, 1]	[**5**, 3, 1]	[**5**, 3, 1]	[**5**, 3, 1]
stride	[**1**, 2, 4]	[**1**, 2, 4]	[**1**, 2, 4]	[**1**, 2, 4]
dimension-1	[16, **64**, 256]	[16, **64**, 256]	[**16**, 64, 256]	[16, **64**, 256]
maxpooling	[**2**, 4]	[**2**, 4]	[**2**, 4]	[**2**, 4]
FC dimension-1	[**128**, 256, 512, 1024]	[**128**, 256, 512, 1024]	[128, 256, 512, **1024**]	[**128**, 256, 512, 1024]
drop ratio-1	[**0.25**, 0.5, 0.75]	[**0.25**, 0.5, 0.75]	[0.25, **0.5**, 0.75]	[**0.25**, 0.5, 0.75]
FC dimension-2	[64, **128**, 256]	[64, **128**, 256]	[64, 128, **256**]	[**64**, 128, 256]
Blk_num-1	[**1**, 4, 8]	[**1**, 4, 8]	[**1**, 4, 8]	[**1**, 4, 8]
dimension-2	[64, 128, **256**]	[64, 128, **256**]	-	-
drop ratio-2	[0.25, **0.5**, 0.75]	[0.25, **0.5**, 0.75]	-	-
Blk_num-2	[**1**, 4, 8]	[**1**, 4, 8]	-	-
Act.Fun.	ReLu			

## Data Availability

Data are available in a publicly accessible repository. The data presented in this study are openly available in IEEE DataPort at 10.21227/7znf-qp86, reference number [60].

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
