# Peer review of "WiTransformer: A Novel Robust Gesture Recognition Sensing Model with WiFi"

_sensors, 2023, doi:10.3390/s23052612_

Round 1

Reviewer 1 Report

The manuscript entitled “WiTransformer: A Novel Robust Gesture Recognition Sensing Model with WiFi” has been investigated in detail. The topic addressed in the manuscript is potentially interesting and the manuscript contains some practical meanings, however, there are some issues which should be addressed by the authors:

1)      In the first place, I would encourage the authors to extend the abstract more with the key results. As it is, the abstract is a little thin and does not quite convey the interesting results that follow in the main paper. The "Abstract" section can be made much more impressive by highlighting your contributions. The contribution of the study should be explained simply and clearly.

2)      The readability and presentation of the study should be further improved. The paper suffers from language problems.

3)            The motivation and contribution should be stated more clearly. Studies on machine learning, if any, of artificial intelligence techniques should be included in the "Introduction" or "Related Works" section.

4)      The importance of the design carried out in this manuscript can be explained better than other important studies published in this field. I recommend the authors to review other recently developed works.

5)      “Evaluation and Discussion” section should be edited in a more highlighting, argumentative way. The author should analysis the reason why the tested results is achieved.

6)           The authors should clearly emphasize the contribution of the study. The feature extraction step should be highlighted. Especially the two-dimensional and three-dimensional feature extraction process seems to be the crucial point of this study. Therefore, the following article can be used to emphasize two-dimensional feature extraction in particular:

The study named- Artificial intelligence-based robust hybrid algorithm design and implementation for real-time detection of plant diseases in agricultural environments- can be used to explain the feature extraction method in the study or to indicate the contribution in the "Section 3 or 5”.

7)      The complexity of the proposed model and the model parameter uncertainty are not enough mentioned.

8)            High complexity architectures such as CNN+LSTM, CNN+GRU, Conv2D and Conv3D were used in the study. Especially in Conv3D, CNN+LSTM and CNN+GRU hybrid architectures, parametric uncertainty should also be considered. The effect of the parametric uncertainty should be discussed.

9)      It will be helpful to the readers if some discussions about insight of the main results are added as Remarks.

This study may be proposed for publication if it is addressed in the specified problems.

Author Response

Special thanks to you for spending much time patiently reviewing our papers despite your busy schedules! Your comments give us great support and guidance regarding writing, experimental rigor, logic, and professionalism and help us better promote related research. Please check attain file.

Reviewer 2 Report

The paper aims at enhancing the identification of spatiotemporal features and improving the robustness for variations of recognition task complexity, For this goal the authors studies two kinds of three-dimensional spatiotemporal feature extractors.

The paper is nice but I have several concerns:

The list of the abbreviations must be sorted.

Figure 1 is unclear. The captions are unreadable and there is no explanation what is the difference between the blue waves and the brown waves.

In the related work section the authors survey several researches. The survey is very interesting and essential. However, the authors cite no paper regarding the extractor even though automatic extractors are more than 20 years old. I would encourage the authors to cite also these two papers: Y. Wiseman & E. Fredj, "Contour Extraction of Compressed JPEG Images", ACM - Journal of Graphic Tools, Vol. 6(3), pp. 37-43, 2001, available online at: https://u.cs.biu.ac.il/~wiseman/jgt2001.pdf  and Dubuisson, M. P., & Jain, A. K., "Contour extraction of moving objects in complex outdoor scenes", International Journal of Computer Vision, Vol. 14(1), pp. 83-105, 1995 which suggests an approach for an extractor.

What is "e" in equation 1?

The authors write in the explanation for equation 2 - e^−j2π f τk(t) is the phase offset caused by the time delay τk(t) on the k-th path. This still makes it unclear what is j and what is f.

I did not succeed to read the captions of Figure 3.

More explanation is needed for equation 4. It is a very complex equation and the authors offer a very slim explanation.

The captions of figure 5, 6 and 7 are also unreadable.

In equation 5, why did the authors choose sqrt(dk) as a scaling factor?

In Figure 14 there is no explanation for the "stairs" in the graph.

The connection between Figure 19 and Table 4 is unclear. Please provide more explanations.

Author Response

Special thanks to you for spending much time patiently reviewing our papers despite your busy schedules! Your comments give us great lessons and guidance from the perspective of writing, experiments, logic, and thinking. We have corrected the errors, reorganized the architecture, and rewritten the unclear contents. Finally, we considered your comments as eleven key points to respond to and presented our revision in the revised version. Please check attain file.

Reviewer 3 Report

In this paper, two transformer backbone models were designed to detect human activities on Wi-Fi signals. This made the detection of human activities feasible in complex scenarios. Evaluation results demonstrated the effectiveness of the proposed method.

The scientific work is sound. However, presentation of the manuscript needs significant improvement. There are numerous English errors. Acronyms and variables used in the equations were not well defined. Sometimes a single variable was used for two different items such as H, which will make the manuscript confusing and it is very difficult to follow.  

Author Response

Special thanks to you for spending much time patiently reviewing our paper despite your busy schedules! Your suggestions and opinions give us great support and guidance regarding writing and professionalism and help us better promote related research. We considered your criticism as four comments to respond to and presented our revision in the revised version. Please check attain file.

Reviewer 4 Report

The actual contribution of the work is human activity recognition (HAR) with WiFi signals ( Robust Gesture Recognition Sensing Model with WiFi)

1.       The contribution should be explained better.  What are the features in the dataset TDSs-22, which are used?  Is the data labelled? How the united spatiotemporal encoder (UST) can extract three-dimensional features only with a one-dimensional encoder.

2.       Much of the useful information about the research is missing. What artifacts from this research can be used by others?

  1. A lot of work have already been employed in this domain. But in "Related Work" section, authors presented limited existing works. It is recommended to present more related work and also a comparative analysis (in terms of a table) so that reader can be convinced by presented results of this research work.
  2. Clearly elaborate on the reason why your proposed method is outperforming others. It is necessary to expand the justification of the variables to be evaluated .Expand the justification of the architecture used (spatiotemporal feature extractors, spatiotemporal feature fusion method) and 3D time-series marking strategy.
  3.  Expand on the rationale or how WI Transformer was determined to be the best among other .

Author Response

To begin with, we thank you for your valuable, positive, and constructive comments. Special thanks to you for spending much time patiently reviewing our papers despite your busy schedules! Your comments give us great support and guidance regarding writing, experimental rigor, logic, and professionalism and help us better promote related research. Please check attain file.

Round 2

Reviewer 1 Report

All my comments have been thoroughly addressed. It is acceptable in the present form.

Reviewer 2 Report

The authors made a decent effort and the paper is certainly publishable so I would recommend accepting the paper.